# Pre-Therapeutic Measurements of Iodine Avidity in Papillary and Poorly Differentiated Thyroid Cancer Reveal Associations with Thyroglobulin Expression, Histological Variants and Ki-67 Index

**DOI:** 10.3390/cancers13143627

**Published:** 2021-07-20

**Authors:** Joachim N. Nilsson, Jonathan Siikanen, Christel Hedman, C. Christofer Juhlin, Catharina Ihre Lundgren

**Affiliations:** 1Department of Molecular Medicine and Surgery, Karolinska Institutet, 17177 Stockholm, Sweden; christel.hedman@ki.se (C.H.); cia.ihre-lundgren@ki.se (C.I.L.); 2Department of Medical Radiation Physics and Nuclear Medicine, Karolinska University Hospital, 17176 Stockholm, Sweden; jonathan.siikanen@sll.se; 3Department of Oncology-Pathology, Karolinska Institutet, 17164 Stockholm, Sweden; christofer.juhlin@ki.se; 4Research and Development Department, Stockholms Sjukhem Foundation, 11235 Stockholm, Sweden; 5Department of Pathology and Cytology, Karolinska University Hospital, 17176 Stockholm, Sweden; 6Department of Breast, Endocrine Tumours and Sarcoma, Karolinska University Hospital, 17176 Stockholm, Sweden

**Keywords:** radioiodine therapy, iodine avidity, differentiated thyroid cancer, papillary thyroid cancer, poorly differentiated thyroid cancer

## Abstract

**Simple Summary:**

In the treatment of thyroid cancer, tumour uptake of iodine is important because it enables the use of radioactive iodine to kill cancer tissue. How radioactive iodine treatments are performed is usually determined by the size of the tumour. This study attempted to find new ways of predicting iodine uptake in advance of treatment. Several factors were found to predict uptake better than tumour size, all relatively simple to investigate in routine healthcare. This information can potentially be used to treat thyroid cancer patients more in accordance with how their tumours behave.

**Abstract:**

Papillary thyroid cancer (PTC) and poorly differentiated thyroid cancer (PDTC) are treated with radioiodine to reduce recurrence and to treat the spread of disease. Adequate iodine accumulation in cancer tissue, iodine avidity, is important for treatment effect. This study investigated which clinical and histological tumour characteristics correlate with avidity. To quantify avidity in cancer tissue, tracer amounts of iodine-131 were given to 45 patients with cytologically confirmed thyroid cancer. At pathology grossing, representative samples of tumour and lymph nodes were taken and subjected to radioactivity quantification ex vivo to determine avidity. Afterwards, samples underwent extended pathology work-up and analysis. We found that tumoural Tg expression and Ki-67 index were correlated with avidity, whereas tumour size and pT stage were not. The histological variant of thyroid cancer was also correlated with iodine avidity. Variants associated with worse clinical prognoses displayed lower avidity than variants with better prognoses. This work provides new information on which tumours have low iodine avidity. Lower avidity in aggressive histological PTC variants may explain their overall poorer prognoses. Our findings also suggest that radioiodine dosage could be adapted to Tg expression, Ki-67 index or histological variant instead of pT stage, potentially improving the efficacy of radioiodine therapy.

## 1. Introduction

Patients with papillary thyroid cancer (PTC) are recommended treatment with surgical resection of one or both lobes of the thyroid gland along with any detectable lymph node metastases. After surgical treatment with total thyroidectomy, radioiodine therapy is given with the aim of ablating thyroid remnants, treating any undetected microscopic disease and, if present, providing treatment of distant metastases [1,2]. In patients with poorly differentiated thyroid cancer (PDTC), a different entity from PTC, treatment is generally similarly initiated with total thyroidectomy and subsequent radioiodine therapy, potentially repeated if the tumour tissue retains the ability to concentrate iodine. The overall outcome of patients with PDTC is worse than with PTC, and commonly used more aggressive treatment modalities for PDTC include external beam radiotherapy and tyrosine kinase inhibitors. The success of radioiodine therapy requires uptake and retention of iodine in the cancer tissue, often summarised as iodine avidity. The mechanisms of iodine uptake, incorporation and retention are altered in various ways in thyroid cancer tissue [3], resulting, for example, in thyroglobulin (Tg) frequently being observed in lower concentrations in thyroid cancer tissue compared to healthy tissue [4]. Furthermore, the expression of plasma membrane sodium/iodide symporter (NIS) and the activity of thyroid peroxidase are generally lower in thyroid cancer compared to healthy tissue [5,6]. In general, patients with iodine-avid disease have a better prognosis than patients with iodine-refractory disease [7,8]. In addition, radioiodine treatment in patients with iodine-avid metastatic disease is known to be associated with improved survival, as shown in published work assessing iodine avidity by using post-therapeutic whole-body scintigraphic scans [9,10,11,12]. In most published studies investigating iodine avidity, it is treated as a binary parameter (i.e., avid/non-avid or positive/negative) derived from nuclear imaging, assessed by the reviewing physician. Using such classifications, strong links have been found for a connection between genetic markers, for example, *BRAF*^V600E^ and *TERT* promoter mutations, and radioiodine refractoriness in thyroid cancer [13]. There is little published quantitative data detailing the range and variations of iodine avidity in thyroid cancer pre-therapeutically. The aim of this study was to provide a more extensive quantitative mapping of the range of iodine avidity in PTC and PDTC and to compare avidity between different histological variants of PTC. Such data have not, to the authors’ knowledge, been published previously and would increase knowledge about which variants of PTC can be expected to respond well to radioiodine therapy. The current work also investigated whether tumour iodine avidity was predicted by Tg expression, tissue proliferation or the parameters recommended to guide the activity choice in published guidelines [2,14,15]. Improving the pre-therapeutic knowledge about which tumours can be effectively treated with radioiodine will enable more individually adapted treatment of thyroid cancer, with potential benefits in terms of improved success rates and fewer unnecessary treatments.

## 2. Materials and Methods

### 2.1. Study Conduct and Patient Inclusion

Patients with PTC preoperatively confirmed through fine needle aspiration cytology between February 2019 and April 2021 were informed and queried for participation in connection to their referral visit to Karolinska University Hospital, Stockholm, Sweden. Patients were included based on the cytological diagnosis and stayed enrolled if the subsequent histopathological analysis determined the diagnosis as PDTC. Follicular thyroid cancers were not included in this study since accurate diagnosis of such entities cannot be done using fine needle aspiration cytology alone, but requires histopathological analysis of the full lesion to differentiate it from follicular adenomas. Adults of all ages were included. Exclusion criteria were pregnancy, severe renal impairment (eGFR < 30 mL/min/m^2^) and difficulties adequately understanding the study information. If the primary tumour size did not allow data collection at grossing without risk of compromising the subsequent histopathological diagnosis, the patient was excluded. The tissue specimens were collected between March 2019 and May 2021.

### 2.2. Patient Specimen Collection and Measurements

Enrolled patients were instructed to keep an iodine-restricted diet for one week prior to an intravenous injection with 5–10 MBq of [^131^I]NaI (GE Healthcare), which was given two days prior to surgery. The delay of two days was chosen to allow for long uptake phases in tumour tissue but not allow decay or wash-out to have a large impact on measurements. A first blood sample was collected a few minutes post-injection, and a second sample was collected during surgery, both used to estimate iodine concentration in blood. The small amount of I-131 given in this study was not deemed to affect the following treatment or prognosis of the thyroid cancer nor cause a significant radiation risk to the participants. The study design was approved by the Radiation Protection Committee at Karolinska University Hospital.

After surgical removal of the thyroid, primary tumour and regional lymph nodes (if indicated), all tissue specimens were sent to pathology grossing. An experienced surgical pathologist or pathology assistant analysed the tissues and cut representative segments from the primary tumour, normal thyroid tissue and, if present, macroscopically suspected lymph node metastases. The normal tissue was acquired from macroscopically normal thyroid tissue from the contralateral lobe if possible and only from the ipsilateral lobe if a lobectomy was performed. In some cases, multiple samples from the primary tumour or lymph node metastases were taken from the same patient to provide information on intra-tumoural variations in avidity. In the case of multifocal cancer in the thyroid gland, a sample from the largest lesion was selected for analysis. Thyroid and tumour samples were handled with separate sterile equipment and all tumour samples were washed with saline solution to avoid cross-contamination of radioactivity.

The tissue and blood samples were weighed and then subjected to ex vivo I-131 activity quantification in a NaI(Tl)-scintillator well chamber (Wallac, 1480 Wizard 3"). The instrument had a crystal with a 3 inch diameter and 3.15 inch height coupled to a linear multi-channel analyser with 1024 channels. The instrument sensitivity was calibrated with known activities of iodine-131 and subsequently normalised and tuned for the appropriate energy window. The energy window was 364 keV +/− 20% and the minimal detectable activity of the system was determined as 0.17 Bq iodine-131, calculated according to the method described by Blue et al. [16]. The dwell time was 10 or 60 min for each sample, depending on the count rate. The geometric effect of the different sample sizes (range: 0.05–5 g) measured with the instrument was determined ahead of study initialisation and correction factors were calculated. The background signal was acquired after each dataset and subtracted from the net counts of each sample. Dead time was corrected for automatically by the instrument software, which was found to accurately correct for samples up to 50 kBq. Above this activity, a further correction was applied, based on separate measurements of the dead time performance (five healthy thyroid samples exceeded 50 kBq, and the highest dead time estimated in the dataset was 35%, requiring a correction of +13%).

After quantification, all samples were immediately returned to the pathology lab for fixation and pathology workup (estimated mean time from surgery to fixation: 3 h), which made it possible for the pathologist to perform histological tissue representation assessment of each unique sample quantified for iodine-131. Laboratory tests were taken pre-operatively for serum TSH and estimated glomerular filtration rate (eGFR).

### 2.3. Histopathology Work-Up

All tumours were grossed and subsequently diagnosed by an experienced endocrine pathologist using histopathology and criteria laid down by the 2017 WHO classification of endocrine tumours [17]. For PTCs, the diagnosis was based on the findings of classical PTC-associated nuclear features. For PDTCs, we used the so-called Turin criteria, according to which solid/trabecular/insular growth was predominant and the presence of >3 mitoses/10 high power fields or necrosis were reported. The pathology report of each specimen contained the diagnosis and the histological variant, as well as a pTNM staging according to the AJCC version 8 criteria [18]. Moreover, all tumours were analysed for Tg and Ki-67 expression using immunohistochemistry. Immunoreactivity was scored for both primary tumours as well as for the largest lymph node metastasis, as the proportion of positive cells (Tg), and by counting a proliferation index in 2000 cells in hotspots (Ki-67). Both immunohistochemical markers are routinely used in our clinical practice, and they were stained using standardised protocols and a Ventana automated slide stainer methodology (Ventana/Roche Diagnostics, Basel, Switzerland) at the Department of Pathology and Cytology, Karolinska University Hospital. For Ki-67, the CONFIRM anti-Ki-67 (clone 30-9) rabbit monoclonal antibody (2 micrograms/mL) was used (Ventana/Roche Diagnostics). Positive external controls mounted on the same slide consisted of human lymph node tissue, and normal colon, kidney and pancreas tissues were also used as references (expecting scattered positivity only). For thyroglobulin, the 2H11 + 6E1 mouse monoclonal antibody (0.33 micrograms/mL) was used (Ventana/Roche Diagnostics), with normal thyroid tissue serving as both external control (unrelated patient) as well as internal control (adjacent to the tumour tissue). For negative controls, lymph node, colon and kidney tissues were used. All control samples were de-identified tissues without the capacity to be traced back to individual patients.

### 2.4. Data Processing

The avidity of tissue was estimated by the normalised iodine concentration at the time of surgery. Normalised concentration, the proportion of injected activity per gram tissue [IA g^−1^], was calculated for all tissue specimen and blood samples and was the estimator used for iodine avidity in tissue. Decay corrections for elapsed time between injection, surgery and quantification were applied, with the time of surgery as reference time. The concentrations in tumour segments and lymph node metastases were corrected for competing uptake in the normal thyroid tissue. This correction was done so that all cancer tissue samples could be compared with equal amounts of iodine available, mimicking the therapeutic radioiodine situation, in which the thyroid gland would be removed. The correction was done according to Equation (1):(1)$$C_{corr}=\frac{C_{meas}}{1-C_{thy}\cdot m_{thy}/A_{inj}}$$
where *C_corr_* (Bq/g) is the corrected activity concentration, *C_meas_* (Bq/g) is the measured activity concentration in a primary tumour or lymph node segment, *C_thy_* (Bq/g) is the activity concentration in the normal thyroid segment, *m_thy_* (g) is the mass of the healthy thyroid, and *A_inj_* (Bq) is the activity injected. When not possible to measure it directly, *m_thy_* was calculated by subtracting the estimated mass of the primary tumour, by volume and density approximation, from the total gland mass.

Iodine clearance from the blood was estimated from the two blood samples and expressed as a fraction of the two concentrations, both decay-corrected to the time of surgery.

The sample purity (the percentage of the sample consisting of tumour, stroma, healthy thyroid tissue, etc.) was assessed by the reviewing pathologist using light microscopy. Using this procedure, we could achieve high sample integrity, and any contaminant thyroid tissue in the primary tumour samples was noted by the pathologist in the study documentation. This allowed corrections for any activity contributions of thyroid tissue by subtraction and by disregarding the mass contribution of any stromal component in the activity concentration calculations.

### 2.5. Statistical Analysis

All statistical analysis was performed using R Version 3.6.3 (R-project.org). Power calculations were performed ahead of study initiation, estimating the number of samples required to find a two-fold difference in iodine avidity in tumours with certain characteristics (assumed to be present in only 10% of patients) with β = 0.1 and α = 0.05. The calculations suggested a sample size of 42 to fulfil the power criteria. Differences in iodine avidity between dichotomous groups were tested using Welch’s *t*-test. If multiple samples from the same subject were available, their geometric means were used for linear regressions and for analyses between dichotomous groups. Linear relationships were found by regression using least mean squares techniques. Correlations were estimated with Pearson’s product–moment correlation coefficient ρ. The data on iodine avidity in the cohort was found to approximately follow a log-normal distribution, assessed through observation and by using Shapiro-Wilk’s test on log-transformed data. Therefore, the geometric mean was used as a descriptive statistic in many of the analyses, as it is robust to small sample sizes and capable of estimating the median of the log-normal distribution.

## 3. Results

Reliable measurements were obtained from 35 patients out of 45 participants. A summary of patient characteristics is shown in Table 1.

In total, 38 samples from primary tumours and 36 samples from lymph node metastases were collected and analysed. Causes for study drop-out were: incorrect cytological diagnosis ahead of surgery (4), primary tumour too small to examine or dissect at grossing (3) and an overly long delay between radioactivity injection and surgery (3). The tumour characteristics are summarised in Table 2.

The measurements showed a large variation of iodine avidity in primary tumours and lymph node metastases, ranging over several orders of magnitude between 6 × 10^−7^ to 2 × 10^−3^ IA g^−1^, as shown in Figure 1.

The iodine avidity in primary versus metastatic lesions from the same patient was also compared. The median fraction between iodine concentration in metastases and their respective primary tumours *C_met_*/*C_primary_* was 0.17 (range: 0.003–10.0). Six out of ten patients had lower avidity in metastases than in primary tumours, for three out of ten it was higher and in one out of ten it was equal.

There was no significant correlation between iodine concentrations in healthy thyroid and primary tumour samples, with a correlation coefficient ρ = −0.09 (95% CI −0.45–0.29).

### 3.1. Histological Variants

To compare the avidity between histological variants, PTC samples were compared in an ”unfavourable” group, based in part on risk estimates from previous research by Ho et al. (hobnail variant, tall cell variant and oxyphilic variant of PTC), and in a group of the other “favourable” variants (conventional, diffuse sclerosing, follicular variant and Warthin-like PTC) [19]. The mean avidity in the unfavourable variant group was calculated as 10.4 (95% CI 2.67–40.4) times lower than the favourable variant group, as shown in Figure 2.

Six samples of PDTC from three patients were analysed, displaying varying avidity across the whole span of the PTC samples. Two samples from the same PDTC tumour exhibited the highest Ki-67 indices in the dataset, but otherwise the characteristics of the PDTC samples aligned well with the unfavourable PTC variants.

### 3.2. Thyroglobulin Expression

The percentage of cells exhibiting immunohistochemical staining of Tg was found to be strongly correlated to iodine avidity, as displayed in Figure 3a. The correlation coefficient between the logarithms of the degree of thyroglobulin staining and the iodine avidity was estimated as ρ = 0.50 (95% CI 0.22–0.70). This relationship was still evident when analysing primary tumours and lymph node metastases separately (ρ = 0.55 and ρ = 0.40, respectively).

### 3.3. Ki-67 Index

Iodine avidity was also found to be strongly inversely correlated to tissue proliferation, estimated by the Ki-67 index, as shown in Figure 3b. Taking the logarithm of both variables, the correlation coefficient was found to be ρ = −0.49 (95% CI −0.70–−0.21). When separated in the analysis, both primary tumours and lymph node metastases had a significant correlation (ρ = −0.57 and ρ = −0.46, respectively).

### 3.4. Age

High age was correlated to lower iodine avidity in tumours and metastases, with a correlation coefficient of ρ = −0.35 (95% CI −0.52–−0.06). The trend was less pronounced than those found for Tg expression or the Ki-67 index.

### 3.5. Tumour Size

There was no significant correlation between iodine avidity and the size of the primary tumour (95% CI −0.52–0.18), as shown in Figure 4a. No significant difference was found (95% CI 0.00016–48) when comparing lower pT stage tumours (pT1a/b + pT2) with higher ones (pT3a/b + pT4a/b). Although non-significant, the difference in avidity between pT stages was large and, upon further analysis, the difference was found to be caused by the high avidity of pT1 tumours, as can be observed in Figure 4b. When comparing pT1a/b versus pT2 + pT3a/b + pT4a/b, higher avidity by a factor of 620 (95% CI 27–14000) was found in the pT1 group.

### 3.6. Blood Iodine Kinetics

Blood concentration of iodine at the time of surgery was not found to be significantly correlated with higher iodine avidity in tumour or metastases ρ = −0.14 (95% CI −0.42–0.16). No significant correlation was found between avidity in cancer tissue and the ratio of blood iodine concentration at surgery over the concentration after injection ρ = −0.10 (95% CI −0.38–0.20) or for eGFR ρ = 0.08 (95% CI −0.22–0.37).

### 3.7. Thyroid Status

Two patients (6% of the total) had lower serum TSH than reference levels and one patient had higher levels (3% of the total) prior to surgery (two-sided 95% confidence interval). All three patients were clinically euthyroid and were included in the analysis.

## 4. Discussion

This study found that the histological variant of PTC, the Ki-67 index and Tg expression all correlate with in vivo iodine avidity. The avidity was quantified with high precision in representative surgical samples of tumour and metastatic tissue. This provides new information on the degree of iodine avidity in papillary and poorly differentiated thyroid cancer. The variation of avidity within the same primary tumour, and between metastases from the same patient, was low compared to differences between patients.

To our knowledge, detailed data on iodine avidity for the different histological variants of PTC described in the WHO 2017 classification have not been published previously. Nakanishi et al. studied the avidity in metastases of conventional PTC, follicular thyroid cancer and follicular variant PTC and grouped patients as having uptake or no uptake on post-therapeutic scans [20]. Their results showed that conventional PTC had worse iodine avidity than the other variants. In the present work, surgical samples were grouped and analysed according to the histological variants of PTC recently shown in a large dataset by Ho et al. to have better or worse prognosis (hobnail variant, tall cell variant and oxyphilic variant of PTC) [19]. The group associated with worse prognosis displayed lower iodine avidity compared to the favourable variants in our dataset. These results are in line with previous work showing a high likelihood for radioiodine refractory disease to be related to the tall cell or hobnail variants of PTC or to PDTC [21,22,23]. This suggests that the differences in iodine avidity observed in the present work may predict the treatment outcome of radioiodine therapy. Most published studies so far have used post-therapeutic imaging to estimate iodine avidity; for example, in work linking high age [24] or *BRAF*^V600E^ mutation [25] to lower iodine avidity. In contrast, the scintillation gamma-counter technique used in this work has a higher sensitivity to radioactivity and can therefore produce more detailed results.

In the adjuvant situation, the radioiodine treatment decision is generally based on analysis of the excised tumour and/or lymph node metastases. We therefore explored the extent to which the determinant parameters in treatment guidelines predict the avidity in tissue. In our material, the pT stage did not predict iodine avidity well. The pT1a/b tumours had higher avidity than others, but there was no significant difference between higher pT stages. Instead, Tg expression, Ki-67 index and histological sub-classification all provided more reliable predictions, suggesting that this information could potentially be useful in clinical routine when selecting radioiodine activity. Given the higher risk of recurrence and the lower avidity in the unfavourable variant group, it could be argued that a higher activity of radioiodine would be best used in this group to decrease the risk of recurrence. It is also possible that this group would benefit more from novel redifferentiation drugs ahead of radioiodine therapy, such as MEK and BRAF inhibitors, which have shown an ability to restore and increase iodine avidity in thyroid cancer [26,27,28].

Thyroglobulin expression assessed by immunohistochemistry has been used for its diagnostic value in differentiating thyroid cancer tissue for decades [29,30]. Dralle et al. reported no difference in radioiodine uptake positivity in recurrent lesions with or without Tg staining; however, they used less sensitive techniques for radioactivity quantification than in the current work [31]. Ringel et al. published results linking reduced Tg mRNA expression with lower NIS mRNA expression but not with iodine uptake in itself [32]. Recent data have shown links between immunohistochemical expression of Tg and risk of disease-specific death in PDTC [33]. The current study suggests the potential use of Tg expression as a predictive marker of iodine avidity, in complement to the body of evidence for its various other uses. The Ki-67 index has previously found use in risk stratification of thyroid cancers, even though it is currently not used in any major staging system [34]. The Ki-67 index has some limitations in that heterogeneous tumours and necrosis can give false negative results, which the current work tried to mitigate by subjecting each measured sample to tissue representation assessment. In addition to the use of risk stratification, we found a clear association between low Ki-67 indices and high iodine avidity in tumour tissue.

The optimal amount of radioactivity in the adjuvant setting or for treatment of distant metastases is still largely unknown and a subject of debate [35,36]. The optimal way of determining iodine avidity and treatment effect would be by performing pre-therapeutic dosimetry using a nuclear imaging modality. This is, however, seldom feasible in the adjuvant setting, where the targets can be expected to be indeterminably small and difficult to detect using nuclear imaging. In the cases of loco-regional residual disease or manifest metastatic disease, image-based dosimetry remains an option, but usage is low and varies between different centres [37]. Determining the targets can also be challenging with remnant thyroid tissue present on the neck. Instead, fixed activities are a commonly used option due to their simplicity, but the information used in the choice of activity should be carefully selected to maximise the benefit-to-side-effect ratio, since ultimately, radioiodine treatment is limited by normal tissue tolerance for short- and long-term side effects [38].

This study was performed without TSH stimulation prior to radioactive iodine administration. TSH stimulation by recombinant human TSH was considered in the study design, but it was not deemed ethical due to possible side effects associated with strongly elevated TSH in patients with a partially normally functioning thyroid gland. Previous research has shown that not all tumours respond to TSH stimulation, but that uptake can be substantially increased for many [39]. It is likely that exogenous TSH stimulation would similarly have increased the iodine concentration in tumour tissue in our experiments. However, the data in the current study were acquired from patients with mostly normal TSH levels, and while they therefore cannot be directly compared to the treatment setting, the TSH levels were much higher than would be present in patients with hormone suppression. It was difficult to distinguish patients that will respond to treatment from those that will not in our dataset, as we did not know the exact concentration in %/g under TSH stimulation. Therefore, we refrained from classifying the samples in this work as “avid” or “non-avid”, instead displaying the full range of iodine concentrations observed. The iodine concentration required to achieve a high enough absorbed dose in the target is also elusive, due to both the lack of conclusive dose-effect data for thyroid cancer tissue and the influence of target size and iodine kinetics in determining the absorbed dose with high precision.

This work used iodine concentration in tumour and metastatic tissue at surgery (range: 44 to 54 h after injection) as an estimate for iodine avidity. Since time-resolved data on iodine concentration would have been practically impossible to obtain, the two-day delay was a compromise between a completed uptake phase and the risk of a low signal due to the decay and wash-out of iodine.

This imposes some limitations on interpretations of avidity reported in this work, as a subset of samples might be misrepresented as having different avidity in this work than in reality, due to the single time-point measurement and the absence of very high TSH levels.

The blood iodine concentration ratio between time of injection and surgery did not correlate to iodine concentration in tumour or metastatic tissue. This indicates that, in our dataset, iodine clearance from blood did not substantially impact how much iodine the cancer tissue accumulated.

Iodine avidity in lymph node metastases was generally lower than in their associated primary tumours. Recent whole-exome sequencing data suggest that the mutational profile differs between primary tumours and distant metastases in PTC, but that driver mutations in the MAPK pathway genes are usually shared [40]. There is, however, evidence that thyroglobulin gene (*TG*) mutations are more common in metastases than in primary tumours, which might result in lower iodine avidity in metastases [41].

Others have investigated the impact of genetic characteristics in thyroid cancer and found that *BRAF*^V600E^ mutations, which activate MAPK signalling, correlate to low iodine avidity [42,43]. Yang et al. have shown that distant metastases with mutations in the *TERT* promoter region are much more likely to exhibit poor iodine avidity in a study that quantitatively assessed iodine uptake using nuclear imaging [44]. This finding is in line with previous work on thyroid cancer, in which *TERT* promoter mutations in general were found to predict worse patient outcomes, irrespective of tumour type [45,46]. This could explain some of the variations observed between different patients in our dataset, and a higher mutational burden in metastases may explain the generally lower uptake in metastases compared to primary tumours.

## 5. Conclusions

We have shown that the iodine avidity in PTC and PDTC might be predicted by the histological variant, Tg expression and Ki-67 index, thereby identifying these parameters as cheap and reproducible methods to predict which tumours should be expected to be resistant to radioiodine therapy. Although it is well-known that the above-mentioned parameters influence the outcome of RAI treatment, the association between these factors and in vivo radioiodine avidity has to our knowledge never before been investigated in a similar setting. In addition, we found that the pT stage, which is the determining parameter of recommended radioiodine activity in many guidelines, correlated poorly with iodine avidity, at least with the common cut-off between pT2 and pT3 tumours. To fully make use of these results, more research is needed to clarify how TSH stimulation affects iodine avidity in different variants of thyroid cancer, and how different variants of PTC and PDTC may be treated with redifferentiation and avidity-restoring drugs.

## Figures and Tables

**Figure 1 cancers-13-03627-f001:**
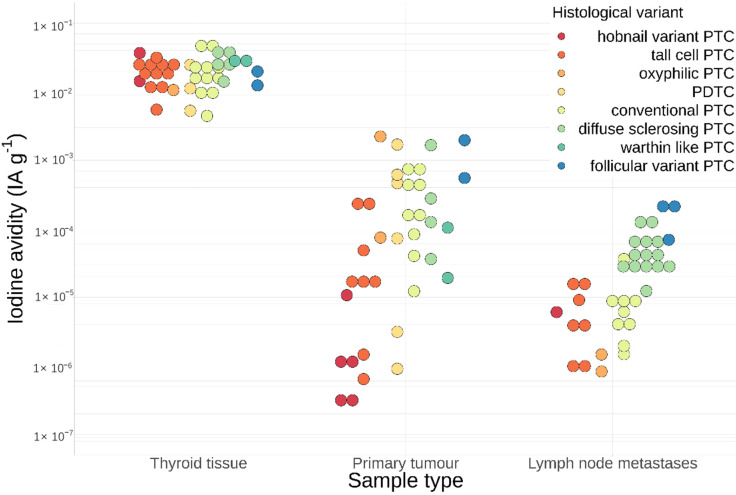
Iodine avidity in samples of healthy thyroid tissue, primary tumour and lymph node metastases. The data are coloured in accordance with the histological variants described by the 2017 WHO classification.

**Figure 2 cancers-13-03627-f002:**
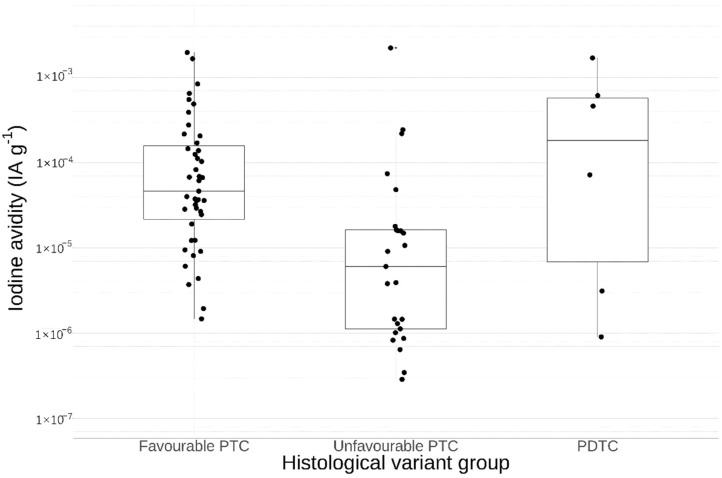
Iodine avidity in favourable (hobnail, tall cell and oxyphilic PTC) versus unfavourable (conventional, follicular variant, diffuse sclerosing and Warthin-like PTC) histological variants, and PDTC. Both primary tumours and metastases are included in this display. The lateral spread in the display is only for illustrative purposes. The factor between the geometric means of the PTC variant groups was calculated as 10.4 (95% CI 2.67–40.4).

**Figure 3 cancers-13-03627-f003:**
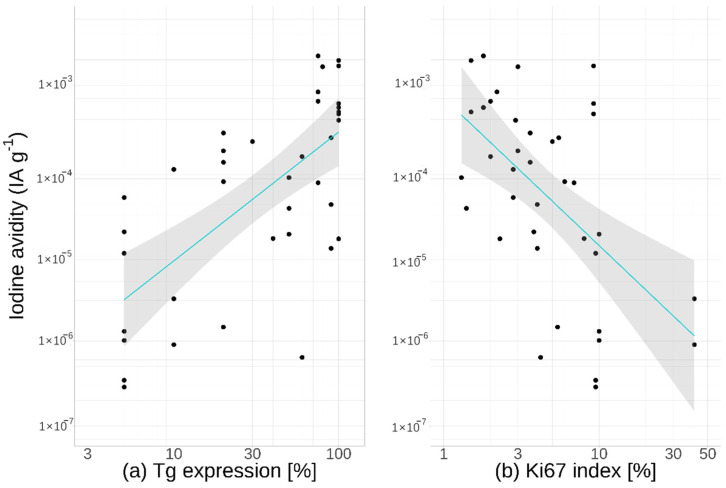
Iodine avidity for all cancer samples versus (**a**) the percentage of cells expressing thyroglobulin and (**b**) the Ki-67 index. Both primary tumours and metastases are included in this display. The line shows the linear fit along with 95% confidence intervals of the fit (shaded area).

**Figure 4 cancers-13-03627-f004:**
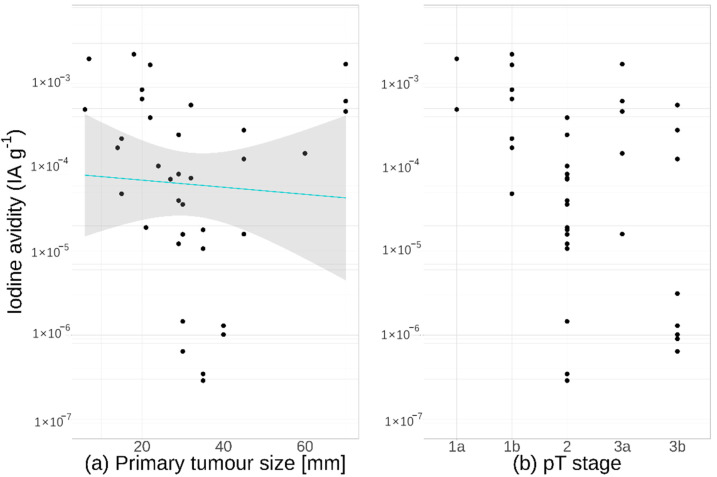
Iodine avidity for all primary tumours. Avidity is plotted against (**a**) tumour size and (**b**) pT stage. The line shows the linear fit along with 95% confidence intervals for the fit (shaded area).

**Table 1 cancers-13-03627-t001:** Clinical characteristics at the time of surgery of patients whose data were included in the analysis. TSH = thyroid stimulating hormone. BMI = body mass index.

Parameter	Total (*n* = 35)
**Age**	
Median (range)	53 (19, 81)
**Sex**	
Male	14 (40%)
Female	21 (60%)
**Time from injection to surgery (h)**	
Median (range)	48 (44, 54)
**BMI (kg/m^2^)**	
Median (range)	26 (19, 34)
**eGFR (ml/min/m^2^)**	
Median (range)	80 (57, 90)
**Serum TSH level (mU/l)**	
High (>4.0)	1 (3%)
Normal (0.4–4.0)	32 (91%)
Low (<0.4)	2 (6%)

**Table 2 cancers-13-03627-t002:** Characteristics of all tumour and metastasis samples collected.

Parameter	Primary Tumour (*n* = 38)	Lymph Node Metastasis (*n* = 36)	Total (*n* = 74)
**Histological type**			
Conventional PTC	9 (24%)	9 (25%)	18 (24%)
Diffuse sclerosing PTC	4 (11%)	14 (39%)	18 (24%)
Warthin-like PTC	2 (5%)	0 (0%)	2 (3%)
Follicular variant PTC	2 (5%)	3 (8%)	5 (7%)
Oxyphilic PTC	2 (5%)	2 (6%)	4 (5%)
Hobnail variant PTC	5 (13%)	1 (3%)	6 (8%)
Tall cell variant PTC	8 (21%)	7 (19%)	15 (20%)
PDTC	6 (16%)	0 (0%)	6 (8%)
**Thyroglobulin expression (%) ***			
Median (range)	50 (5, 100)	60 (5, 100)	50 (5, 100)
**Ki-67 index (%) ***			
Median (range)	4.0 (1.3, 41)	3.5 (1.0, 14)	4.0 (1.0, 41)
**Extrathyroidal/nodal extension**			
Yes/No	10/28 (26%/74%)	10/26 (28%/76%)	20/54 (27%/73%)
**Negative surgical margins ****			
Yes/No	29/9 (76/24%)	36/0 (100/0%)	65/9 (88/12%)
**Iodine avidity (log_10_ of IA g^−1^)**			
Median (range)	−4.1 (−6.5 −2.7)	−4.8 (−6.1 −3.7)	−4.5 (−6.5 −2.7)

* One lymph node metastasis not shown because immunohistochemistry was not feasible (diameter < 1 mm). ** Microscopical margins.

## Data Availability

The data presented in this study are available on request from the corresponding author. The data are not publicly available due to privacy reasons.

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
