# Peer review of "Pre-Therapeutic Measurements of Iodine Avidity in Papillary and Poorly Differentiated Thyroid Cancer Reveal Associations with Thyroglobulin Expression, Histological Variants and Ki-67 Index"

_cancers, 2021, doi:10.3390/cancers13143627_

Round 1
Reviewer 1 Report
Overall, I find the study very interesting, as showing that iodine-avidity actually depends from tumor biology and is independent from tumor size. This may open new approaches for the decision-making of adjuvant post-surgical RAI, in the sense that biological features of the tumor, including the variants and the ki67 (which to date does not have specific role), may overcome tumor dimensions (pT) in the definition of to do or not RAI (in small PTC obviously). Therefore, features predictive of RAI-refractoriness, including genetics as well as the BRAFV600E mutation, may represent the real prognostic factor in DTC and PDTC. These concepts should be stressed by authors in the discussion section
As I will stress later, I don’t understand the absence of FTC, so authors should explain.
A critical point is to check that the nuclear medicine methodology was acceptable and consistent…otherwise results may be misleading. Therefore, I ask for a methodological review, to be performed by a nuclear medicine specialist (I am endocrinologist), to assess this aspect.
- Introduction
- Patients with papillary thyroid cancer (PTC) and poorly differentiated thyroid cancer (PDTC) are recommended treatment with surgical resection of one or both lobes of thyroid gland along with any detectable lymph node metastases.
I suggest to split PTC and PDTC into 2 separate subparagraphs describing epidemiology and main clinical features. These should include the typical clinical presentation and outcome (DFS for PTC and OS for PDTC). It should be stressed that surgical treatment of PTC has to be calibrated based on disease extension, whereas PDTC requires a more radical approach by definition, due to the worst evolution. Afterwards, I should stress heterogeneity of PDTC, which traduces in a variable response to RAI. In conclusion, I would mark that PTC and PDTC are 2 different entities.
- Please better remark genetics highlighting the iodine-avidity, also referring the the role of BRAFV600E. Please cite the review paper Marotta V, Sciammarella C, Colao AA, Faggiano A. Application of molecular biology of differentiated thyroid cancer for clinical prognostication. Endocr Relat Cancer. 2016 Aug 30. pii: ERC-16-0372. [Epub ahead of print] PubMed PMID: 27578827.
- Methods
- I am concerned about the way of diagnosing PTC and PDTC pre-operatively. Actually, the definition “preoperatively confirmed PTC and PDTC” is not correct by definition according to my opinion. Cytology is a very accurate tool, but also limited as based on few groups of cells. It does not allow to catch the solid/trabecular/insular pattern of growth as well as the mitotic activity and the necrosis, which are the Turin diagnostic criteria of PDTC (as reported also in the text). Similarly, FNC does not always allow to distinguish between PTC and other malignancies, mainly FTC, especially for the follicular variant of PTC. Authors should solve this concern.
- Another concern is about the ethics…I see that the study has been submitted and approved by the Regional Ethical Review Board in Stockholm and the Radiation Protection Committee at Karolinska University Hospital, but I ask authors to specify it in the text also explaining that such a radioactive administration, performed without any clinical improvement for the patient, actually does not produce any damage.
- Why authors chose to exclude FTC from the analysis? It was done with a particular rationale or represented an occasional feature of the study? Please clarify
- Have results, namely I-131 accumulation in the thyroid tissue, being normalized for renal function. eGFR ranged from 57 to 90 and this may impact on the analysis. I think that normalizing results for renal function may provide a more accurate result, even though having an eGFR less than 30 was an exclusion criteria and no correlation was found between I-131 tissue avidity and eGFR.
- Crucial: check the methodology with a nuclear medicine revisor
- Results
- When reporting results for Tg and Ki67, please clarify in the text (it is strongly clear in the figure) the sense of the relationship. It is important that readers understand that higher is the avidity higher is the Tg (%expressing cells) and lower is ki67, therefore there is a direct correlation between avidity and Tg (higher grade of differentiation, less aggressive disease) and an inverse correlation between avidity and ki67 (lower grade of differentiation, higher biological activity, worst outcome).
- I ask for analyzing PDTC separately…3 groups. a) favourable PTC variants b) unfavourable PTC variants c) PDTC. PDTC is an independent but heterogeneous entity, and the heterogeneity rely on the variable RAI avidity.
- Discussion
ok
Author Response
Please find enclosed our replies to your review of our manuscript.
Sincerely,
Joachim Nilsson

Author Response

(The authors gave the same response as above.)

Round 2
Reviewer 1 Report
well done accept
Author Response
Thank you.